# Mutual Coupling Reduction of Cross-Dipole Antenna for Base Stations by Using a Neural Network Approach

**Ersin Ozdemir [1], Oguzhan Akgol [1], Fatih Ozkan Alkurt [1], Muharrem Karaaslan [1], Yadgar I. Abdulkarim [2,3] and Lianwen Deng [2,\*]**

[1] Electrical and Electronics Engineering, Iskenderun Technical University, Iskenderun 31200, Hatay, Turkey; ersin.ozdemir@iste.edu.tr (E.O.); oguzhan.akgol@iste.edu.tr (O.A.); fozkan.alkurt@iste.edu.tr (F.O.A.); muharrem.karaaslan@iste.edu.tr (M.K.)

[2] School of Physics and Electronics, Central South University, Changsha 410083, China; yadgar.kharkov@gmail.com

[3] Physics Department, College of Science, University of Sulaimani, Sulaimani 46001, Iraq

\* Correspondence: denglw@csu.edu.cn

**Abstract:** In this manuscript, a resonator layer is presented for the purpose of reducing the mutual coupling effect between each antenna element of a cross dipole antenna. In design processes, an artificial neural network approach was used for various resonator designs. In the operating frequency band of 2.2–2.7 GHz, 48 different $6 \times 6$ resonator layers were created and integrated into the cross dipole antenna to reduce transmission and improve isolation between each antenna elements. Moreover, when training an artificial neural network in the Matlab program, 48 different resonator layers were used with the return losses and transmission values of cross dipole antenna elements. After training process, eight unknown resonator designs were tested and accurate results were obtained. Finally, one of the resonator planes, which was obtained from the artificial neural network, was fabricated and experimentally tested, then an accurate result was obtained. This study provides a good solution, especially for improving isolation in multiport antenna systems, using an artificial neural network approach.

**Keywords:** cross-dipole antenna; artificial neural network; isolation improvement; wireless communication

## 1. Introduction

Mutual coupling effect is a serious problem in Multiple Input-Multiple Output (MIMO) antennas, and this problem occurs if the antenna elements are located very closely to each other. For example, if two planar antennas are located in a close range, one of these antennas radiates relatively as a function of the other antennas and vice versa [1]. In addition, it is expected for the antenna performances to be affected due to the close placement of the antennas. The physical spaces that host antennas (radio enclosures, antenna towers, etc.) are mostly limited, meaning it is inevitable to place multiple antennas and/or antenna elements very close to each other. This condition causes mutual coupling problems, that is, isolation problems. In recent years, various structures were designed and integrated with various antennas to overcome this problem and improve the efficiency of the antenna systems [2–6]. For instance, a wide band neutralization line [2] and meander line resonator [3] were presented in the literature to enhance mutual coupling between each antenna element of MIMO antennas. Moreover, metamaterials [4,5] and metasurfaces [6], which are other artificial structures, were also used to minimize the mutual coupling effect in MIMO antenna applications.

In addition, Electromagnetic Band Gap (EBG) structures are also used in mutual coupling reduction applications [7,8]. Li et al. proposed a two layered EBG structure to reduce mutual coupling effects between ultra-wide band monopole antennas in a wireless communication band [7]. In addition, one dimensional EBG and split ring resonator structures were used to decrease the coupling problem of 2.45 GHz planar monopole antennas [8]. Similarly, Ibrahim et al. proposed a MIMO patch antenna configuration for a 5.8 GHz band in a ground plane to overcome this coupling problem between antenna elements [9]. In another study, Bernety and Yakovlev succeeded to reduce mutual coupling between nearest strip dipole antennas with the help of elliptical metasurface layers [10]. Moreover, Cheng et al. designed an array patch antenna configuration with the integration of an polarization conversion isolator to improve the antenna system parameters distorted by the coupling effect [11]. Additionally, there are many approaches and studies in the related literature focusing on improving the isolation by reducing the coupling effects between closely placed antenna elements in MIMO antennas [12–15]. In one of the recent studies, a metamaterial absorber structure is designed and integrated to a four-element array antenna for 1.27 GHz band by Zhang et al. [14]. With these developed techniques, artificial neural networks provide accurate solutions and are used in many areas [16–20]. Srivastava et al. used artificial neural network for analyzing and estimating microstrip circular patch antenna parameters in S frequency band regime [16]. In addition, Rajaraman et al. developed a method to compute the resonance frequency of metamaterial based patch antennas with the help of a artificial neural network [17]. In the same way, Nayak and Kumar used a artificial neural network approach to design a microstrip patch antenna with possible optimization values [18]. In another study, a novel artificial neural network model was developed to estimate antenna performance parameters in X band antenna applications [19]. As shown in a literature review, various structures were designed and developed for reducing the mutual coupling of cross antennas, while neural network approaches were used to enhance and estimate antenna performance parameters.

In this study, a mutual coupling effect between each antenna element of cross dipole antenna is reduced and the isolation between them is improved by using an artificial neural network model. Firstly, an artificial neural network approach and working principle was given and the design phase of cross dipole antenna was presented for the 2.45 GHz frequency band. To minimize transmission values between antenna elements as well as corresponding mutual coupling, a resonator layer was designed and integrated to the cross dipole antenna. Afterwards, 48 different types of resonators were tested with a cross dipole antenna, the obtained results were transferred to the neural network, and the network was then trained. Moreover, a network model was tested and one of the test resonators was fabricated and experimentally investigated.

## 2. Neural Network Approach

Before mentioning artificial neural networks (ANN), biological neurons should be discussed. Neurons in our brains are connected to each other through synapses, while signals are transmitted between neurons through each neuron cell bonds. The signals are carried to the neurons to which the synapses are bound, and when the sum of these signals exceeds a certain level or activation level, the neuron outputs and transmits to the other neurons with synapses. It is known that there are 100 billion of neurons in our brain. Each of these neurons makes around a thousand connections with each other. These complex connections enable us to achieve many functions such as thinking, linking events, and recording memories.

During the last half century, researchers have been to put forward a mathematical model inspired by the human brain and its features. This model, which is called an artificial neural network, is built on multiple input lines, a connected neuron, and a single output system connected to the same neuron. In the illustration given in Figure 1, each input line has a value called a weight. This weight value is multiplied by the input value and the value of that line is found. All inputs are multiplied by the weights in the line they are located on and by the neuron to which they are connected, and the

threshold value is added. The total value is found and passed through the activation function. Finally, the result is the output value of the neuron.

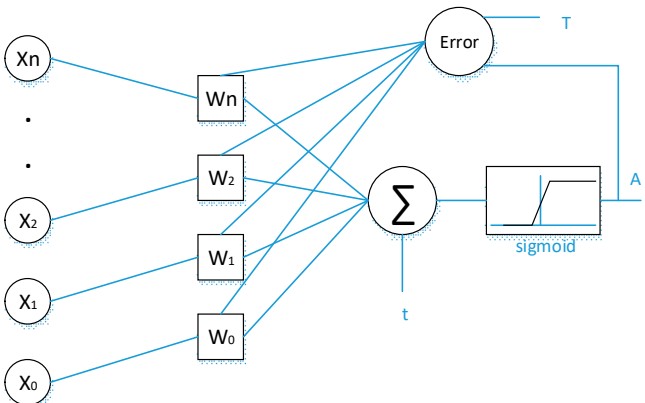

**Figure 1.** Illustration of the neural network approach.

The simple formulas of this structure are given below; in these formulas, *i* input and *w* weight are multiplied with each other, and the obtained results for each input and weight are added together to obtain the value of *s* output. For the Jth neuron,

$$\mathbf{sum_j} = \sum \mathbf{input_i} * \mathbf{weight_{ij}} \tag{1}$$

$$\mathbf{output_j} = \mathbf{f_{sigmoid}}\left(\mathbf{sum_j} + \mathbf{threshold_j}\right). \tag{2}$$

With this sum value (1), the threshold value is added and the output of the neuron is obtained. When this value is passed through the activation function, then the value of that neuron will transmit to other neurons. In this study, the sigmoid function was selected as the activation function.

When a young child learns to read and write, he or she generally must be educated in a class. The child must repeat the letter A several times to learn how to write it. This repetition occurs under the supervision of the teacher and feedback is given to the student about his/her mistakes. The same process could be applied to artificial neural networks. The data to be used in artificial neural networks are compared with the response of the network. As a result of this comparison, the obtained value is the error value. This error value should be close to zero. After the calculation of each error value, the weight values should be updated. This process continues until the error approaches zero. Neurons are traced backwards and their weights are updated according to the error rates. In Figure 1, the inputs are multiplied by the weights and transferred by neurons. The obtained result is compared to the target values and weights are adjusted again. With zero resets, learning is achieved and the training of the network is completed.

The formulas used to update the weights with the back propagation algorithm of the network are given below:

$$\mathbf{weight_{ij}} = \mathbf{weight'_{ij}} + \mathbf{Learning\ rate} * \mathbf{error_j} * \mathbf{input_i} \tag{3}$$

$$\mathbf{error_j} = \mathbf{output_j} * \left(1 - \mathbf{output_j}\right) * \left(\mathbf{target_j} - \mathbf{output_j}\right). \tag{4}$$

The information taught to artificial neural networks is stored in the weight values of the connections of the network. Since the weight values are spread over the entire network, the memory of the network is also called distributed memory.

## 3. Cross-Dipole Antenna Design

In this study, nested dipole antennas were designed as a cross-dipole antenna. In design processes, Finite Integration Technique (FIT) based microwave simulation software is used. Designed cross-dipole

antenna is illustrated with dimensions in Figure 2a. This type of antenna is designed for 2.40 GHz communication frequencies and each pole has a 30.75 mm length. Moreover, FR-4 dielectric material is used in dielectric layers which have a permittivity of 4.3 and copper is used in antenna resonators which have a conductivity of $5.8 \times 10^8$ S/m. Designed cross-dipole antenna has a good return loss of $-17$ dB at 2.45 GHz, as illustrated in Figure 2b. Moreover, there is an undesired transmission between each antenna as shown, and the transmission value is nearly $-15$ dB at 2.45 GHz. The purpose of this study was to reduce this undesired transmission characteristic and improve the isolation between two antenna elements forming the cross dipoles. For this reason, a resonator plane was designed and integrated to the cross dipole antenna, as presented in the next section.

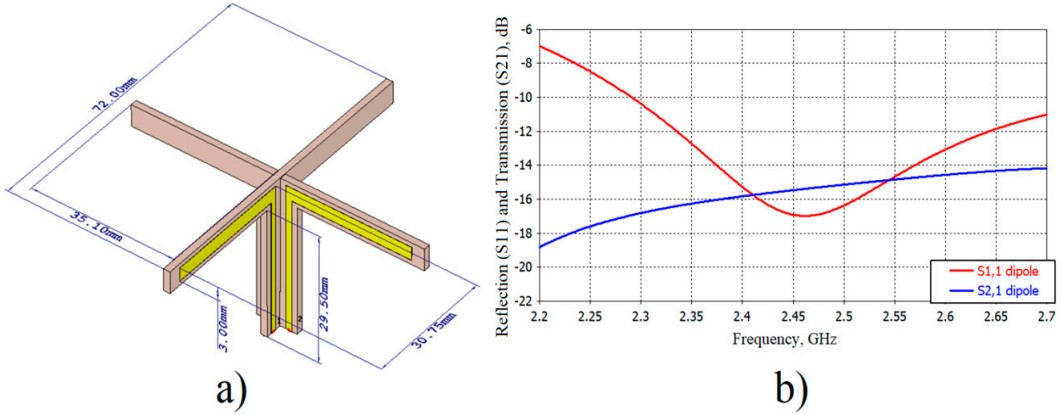

**Figure 2.** (**a**) Cross-dipole antenna design with dimension parameters and (**b**) Simulated results for $S_{11}$, $S_{21}$.

## 4. Reduction of Mutual Coupling by a Neural Network Approach

As mentioned before, the aim of this study was to reduce mutual coupling between each antenna element of a cross dipole antenna. The mutual coupling effect corresponds to the transmission value in antenna engineering and a resonator plane was designed in our study to reduce the transmission value (S21) in the designed cross dipole antenna. As shown in Figure 3, the designed resonator plane has a square shape and 50 mm side lengths, and it also consists of unit cells made of copper. Each unit cell has dimensions of 5 mm × 5 mm and resonator plane consists of 6 × 6 unit cells, as illustrated in Figure 3. According to the number of unit cells, there are 236 different artificial surface possibilities. It is hard to reduce the mutual coupling effect in FIT based microwave simulator because doing so involves the trial and error method. However, an artificial neural network model gives more accurate results with a trained data set. Thus, we used an artificial neural network model.

In line of the purpose of our study, an arbitrary resonator plane was designed on a printed circuit board and integrated into the cross dipole antenna to reduce the mutual coupling effect. Each unit cell is represented by a binary bit "1" or "0" state. Therefore, the "1" state corresponded to a copper cell while "0" corresponded to an empty state. Although there were 36 bits, the designed surface was represented by 18 bits to set up a symmetric design procedure, as illustrated in Figure 4. Each symmetric column is named as X, Y, or Z for identification of the relevant cell number. A total of 48 various resonator planes were designed and tested with cross dipole antenna. The obtained return losses and transmission values are listed in Table 1.

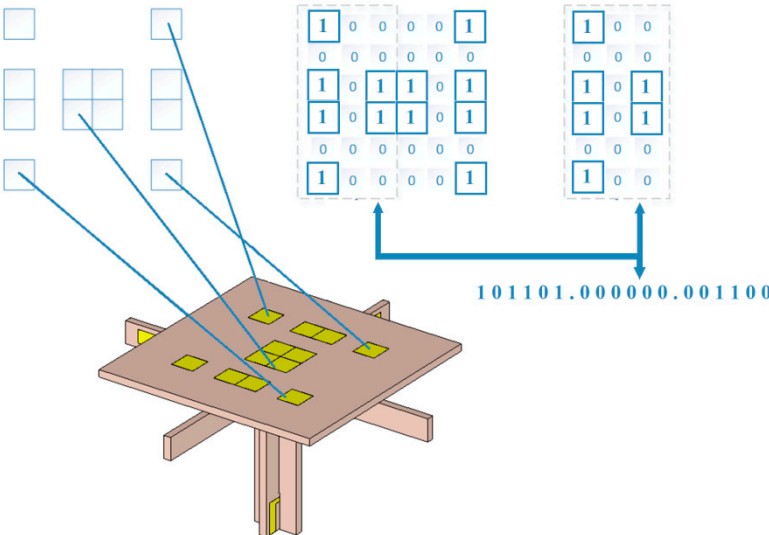

**Figure 3.** Cross-dipole antenna and resonator plane in a binary system.

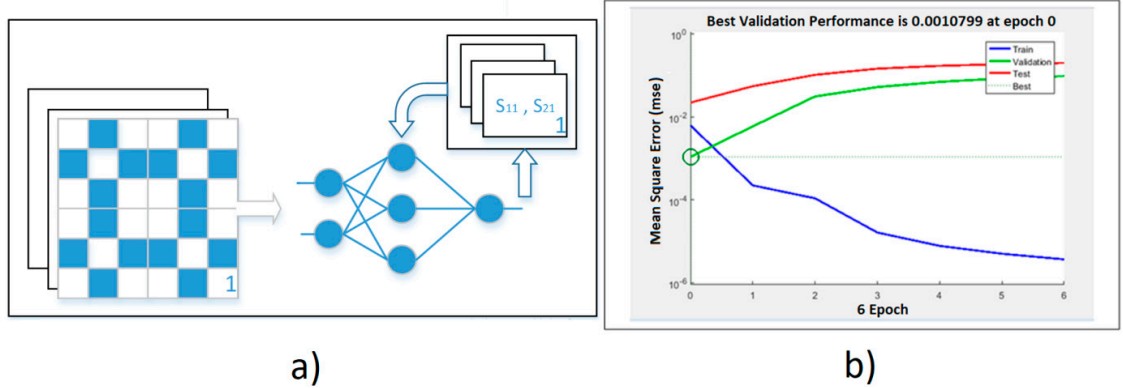

**Figure 4.** (**a**) Resonator plane and artificial neural network working principle and (**b**) neuron training results.

These return loss and transmission values were exported to the Matlab environment and the artificial neural network toolbox was used for training processes. One of the relevant designed surface and working processes for artificial neurons is given in Figure 4a. A total of 48 various results were trained by artificial neurons and neural network tools in the Matlab environment, and mean square errors (MSE) were calculated and plotted as shown in Figure 4. As shown in Figure 4b, the artificial neural network was fully trained at the sixth iteration.

Afterwards, the trained neural network was tested by eight different unknown resonator planes, as listed in Table 2. According to the obtained isolation (S21) values, the trained neural network was well matched with the results of the FIT based microwave simulator, as the error rate was less than 2%. For example, the isolation value was obtained as −15.5183 dB for the resonator plane (X, Y, Z) represented by (00000, 000000, 111110) and the isolation value was −15.5 dB in the FIT based simulator. As a consequence, the artificial neural network was well trained by different resonator planes and the obtained results are plotted in Figure 5 with eight unknown test planes. In addition, according to Figure 4, there was a good match in isolation values with a mean error rate of less than 2%, which is listed in Table 2.

**Table 1.** Return loss and transmission values of 48 various resonator planes.

| | | | | | Variables | | | | | | | | | | | | | | Output1 | Output2 | Outpu3 Isolation |
|---|---|---|---|---|---|---|---|---|---|---|---|---|---|---|---|---|---|---|---|---|---|
| X1 | X2 | X3 | X4 | X5 | X6 | Y1 | Y2 | Y3 | Y4 | Y5 | Y6 | Z1 | Z2 | Z3 | Z4 | Z5 | Z6 | S11 | S22 | S12 |
| 0 | 0 | 0 | 0 | 0 | 0 | 0 | 0 | 0 | 0 | 0 | 0 | 0 | 0 | 0 | 0 | 0 | 0 | −20.64 | −14.82 | −14.98 |
| 0 | 0 | 0 | 0 | 0 | 0 | 0 | 0 | 0 | 0 | 0 | 0 | 0 | 1 | 1 | 0 | 0 | 0 | −30.78 | −10.59 | −15.89 |
| 0 | 0 | 0 | 0 | 0 | 0 | 0 | 0 | 0 | 0 | 0 | 0 | 1 | 1 | 1 | 0 | 0 | 0 | −27.15 | −10.56 | −15.90 |
| 0 | 0 | 0 | 0 | 0 | 0 | 0 | 0 | 0 | 0 | 0 | 0 | 1 | 1 | 1 | 1 | 0 | 0 | −13.73 | −9.77 | −15.43 |
| 0 | 0 | 0 | 0 | 0 | 0 | 0 | 0 | 0 | 0 | 0 | 0 | 1 | 1 | 1 | 1 | 1 | 0 | −7.84 | −9.68 | −15.51 |
| 0 | 0 | 0 | 0 | 0 | 0 | 0 | 0 | 0 | 0 | 0 | 0 | 1 | 1 | 1 | 1 | 1 | 1 | −4.64 | −9.67 | −15.86 |
| 0 | 0 | 0 | 0 | 0 | 0 | 1 | 0 | 0 | 0 | 0 | 0 | 1 | 1 | 1 | 1 | 1 | 1 | −3.26 | −9.12 | −16.11 |
| 0 | 0 | 0 | 0 | 0 | 0 | 1 | 1 | 0 | 0 | 0 | 0 | 1 | 1 | 1 | 1 | 1 | 1 | −3.78 | −8.38 | −16.19 |
| 0 | 0 | 0 | 0 | 0 | 0 | 1 | 1 | 1 | 0 | 0 | 0 | 1 | 1 | 1 | 1 | 1 | 1 | −4.78 | −6.74 | −16.51 |
| 0 | 0 | 0 | 0 | 0 | 0 | 1 | 1 | 1 | 1 | 0 | 0 | 1 | 1 | 1 | 1 | 1 | 1 | −5.58 | −6.09 | −16.61 |
| 0 | 0 | 0 | 0 | 0 | 0 | 1 | 1 | 1 | 1 | 1 | 0 | 1 | 1 | 1 | 1 | 1 | 1 | −5.62 | −5.96 | −16.64 |
| 0 | 0 | 0 | 0 | 0 | 0 | 1 | 1 | 1 | 1 | 1 | 1 | 1 | 1 | 1 | 1 | 1 | 1 | −4.96 | −5.86 | −16.80 |
| 1 | 0 | 0 | 0 | 0 | 0 | 1 | 1 | 1 | 1 | 1 | 1 | 1 | 1 | 1 | 1 | 1 | 1 | −4.01 | −5.23 | −16.81 |
| 1 | 1 | 0 | 0 | 0 | 0 | 1 | 1 | 1 | 1 | 1 | 1 | 1 | 1 | 1 | 1 | 1 | 1 | −4.35 | −4.77 | −16.81 |
| 1 | 1 | 1 | 0 | 0 | 0 | 1 | 1 | 1 | 1 | 1 | 1 | 1 | 1 | 1 | 1 | 1 | 1 | −4.94 | −3.97 | −16.88 |
| 1 | 1 | 1 | 1 | 0 | 0 | 1 | 1 | 1 | 1 | 1 | 1 | 1 | 1 | 1 | 1 | 1 | 1 | −5.35 | −3.60 | −16.85 |
| 1 | 1 | 1 | 1 | 1 | 0 | 1 | 1 | 1 | 1 | 1 | 1 | 1 | 1 | 1 | 1 | 1 | 1 | −5.33 | −3.43 | −16.76 |
| 1 | 1 | 1 | 1 | 1 | 1 | 1 | 1 | 1 | 1 | 1 | 1 | 1 | 1 | 1 | 1 | 1 | 1 | −4.90 | −3.24 | −16.70 |
| 1 | 1 | 1 | 1 | 1 | 1 | 1 | 1 | 1 | 1 | 1 | 1 | 1 | 1 | 1 | 0 | 1 | 0 | −4.77 | −3.22 | −15.94 |
| 1 | 1 | 1 | 1 | 1 | 1 | 1 | 1 | 1 | 0 | 1 | 1 | 1 | 1 | 1 | 0 | 1 | 0 | −4.66 | −3.30 | −15.78 |
| 1 | 0 | 1 | 1 | 1 | 1 | 1 | 1 | 1 | 0 | 1 | 1 | 1 | 1 | 0 | 1 | 0 | 1 | −3.87 | −3.40 | −15.49 |
| 1 | 0 | 0 | 1 | 1 | 1 | 1 | 1 | 1 | 0 | 1 | 1 | 1 | 1 | 0 | 1 | 0 | 1 | −3.14 | −3.75 | −15.21 |
| 1 | 0 | 0 | 0 | 1 | 1 | 1 | 1 | 1 | 0 | 1 | 1 | 1 | 1 | 0 | 1 | 0 | 1 | −1.24 | −4.61 | −15.33 |
| 1 | 0 | 0 | 0 | 0 | 1 | 1 | 1 | 1 | 0 | 1 | 1 | 1 | 1 | 0 | 1 | 0 | 1 | −1.24 | −5.00 | −15.77 |
| 1 | 0 | 0 | 0 | 0 | 1 | 1 | 1 | 0 | 0 | 1 | 1 | 1 | 1 | 0 | 1 | 0 | 1 | −19.42 | −5.94 | −16.23 |
| 1 | 0 | 0 | 0 | 0 | 1 | 1 | 0 | 0 | 0 | 1 | 1 | 1 | 1 | 0 | 1 | 0 | 1 | −20.90 | −6.34 | −16.28 |
| 1 | 0 | 0 | 0 | 0 | 1 | 1 | 0 | 0 | 0 | 1 | 1 | 1 | 0 | 0 | 0 | 0 | 1 | −20.00 | −6.43 | −16.60 |
| 1 | 0 | 0 | 0 | 0 | 1 | 1 | 0 | 0 | 0 | 1 | 1 | 1 | 0 | 0 | 0 | 0 | 1 | −19.83 | −6.53 | −16.67 |
| 1 | 0 | 0 | 0 | 0 | 1 | 1 | 0 | 0 | 0 | 1 | 0 | 1 | 0 | 0 | 0 | 0 | 1 | −20.49 | −4.62 | −16.90 |
| 1 | 0 | 0 | 0 | 0 | 1 | 1 | 0 | 0 | 0 | 1 | 1 | 0 | 0 | 0 | 0 | 0 | 1 | −21.16 | −4.88 | −17.02 |
| 1 | 0 | 0 | 0 | 0 | 1 | 0 | 0 | 0 | 0 | 1 | 1 | 0 | 0 | 0 | 0 | 0 | 1 | −21.57 | −4.90 | −17.06 |
| 0 | 0 | 0 | 0 | 0 | 1 | 0 | 0 | 0 | 0 | 1 | 0 | 0 | 0 | 0 | 0 | 0 | 1 | −21.83 | −4.90 | −17.09 |
| 0 | 0 | 0 | 0 | 0 | 1 | 0 | 0 | 0 | 0 | 0 | 0 | 0 | 0 | 0 | 0 | 0 | 1 | −21.27 | −13.09 | −16.27 |
| 1 | 1 | 1 | 0 | 0 | 1 | 0 | 0 | 0 | 0 | 0 | 0 | 0 | 0 | 0 | 0 | 0 | 1 | −21.28 | −13.11 | −16.13 |
| 1 | 1 | 1 | 1 | 1 | 1 | 0 | 0 | 0 | 0 | 0 | 0 | 0 | 0 | 0 | 0 | 0 | 1 | −11 | −12.82 | −16.34 |
| 1 | 1 | 1 | 1 | 1 | 1 | 0 | 0 | 0 | 0 | 0 | 1 | 0 | 0 | 0 | 0 | 0 | 1 | −5.35 | −16.78 | −16.50 |
| 1 | 1 | 1 | 1 | 1 | 1 | 1 | 0 | 0 | 0 | 0 | 1 | 1 | 0 | 0 | 0 | 0 | 1 | −1.9 | −1.95 | −13.31 |
| 1 | 1 | 1 | 1 | 1 | 1 | 1 | 0 | 0 | 0 | 0 | 1 | 0 | 0 | 0 | 0 | 0 | 0 | −5.47 | −12.17 | −16.46 |
| 1 | 1 | 0 | 0 | 1 | 1 | 1 | 0 | 0 | 0 | 0 | 1 | 0 | 0 | 0 | 0 | 0 | 0 | −19.75 | −12.55 | −16.41 |
| 1 | 1 | 0 | 0 | 1 | 1 | 1 | 0 | 0 | 0 | 0 | 1 | 0 | 0 | 1 | 1 | 0 | 0 | −24.53 | −9.74 | −15.79 |
| 1 | 1 | 0 | 0 | 1 | 1 | 1 | 0 | 0 | 0 | 1 | 0 | 0 | 0 | 1 | 1 | 0 | 0 | −1.57 | −1.22 | −10.45 |
| 0 | 1 | 0 | 0 | 1 | 0 | 1 | 0 | 0 | 0 | 1 | 1 | 0 | 0 | 1 | 1 | 0 | 0 | −25.10 | −9.73 | −15.75 |
| 0 | 1 | 1 | 0 | 1 | 0 | 1 | 0 | 0 | 0 | 1 | 1 | 0 | 0 | 1 | 1 | 0 | 0 | −20.95 | −9.79 | −15.71 |
| 0 | 1 | 1 | 1 | 1 | 0 | 1 | 0 | 0 | 0 | 1 | 1 | 0 | 0 | 1 | 1 | 0 | 0 | −7.63 | −9.73 | −16.26 |
| 0 | 1 | 1 | 1 | 1 | 0 | 1 | 0 | 1 | 0 | 1 | 1 | 0 | 0 | 1 | 1 | 0 | 0 | −7.76 | −7.11 | −15.20 |
| 0 | 1 | 1 | 1 | 1 | 0 | 1 | 0 | 1 | 0 | 1 | 1 | 0 | 0 | 1 | 1 | 0 | 0 | −8.27 | −1.64 | −14.89 |
| 0 | 0 | 1 | 1 | 0 | 0 | 0 | 0 | 0 | 0 | 0 | 0 | 0 | 0 | 1 | 1 | 0 | 0 | −27.53 | −9.88 | −15.59 |
| 1 | 0 | 1 | 1 | 0 | 1 | 0 | 0 | 0 | 0 | 0 | 0 | 0 | 0 | 1 | 1 | 0 | 0 | −27.01 | −9.87 | −15.63 |

**Table 2.** Comparison of transmission values of eight unknown resonator planes.

| X1 | X2 | X3 | X4 | X5 | X6 | Y1 | Y2 | Y3 | Y4 | Y5 | Y6 | Z1 | Z2 | Z3 | Z4 | Z5 | Z6 | Artificial Neural Network | FIT Based Simulator | % Error |
|---|---|---|---|---|---|---|---|---|---|---|---|---|---|---|---|---|---|---|---|---|
| 0 | 0 | 0 | 0 | 0 | 0 | 0 | 0 | 0 | 0 | 0 | 0 | 0 | 0 | 0 | 0 | 0 | 0 | −15.2782 | −15 | 1.85 |
| 0 | 0 | 0 | 0 | 0 | 0 | 0 | 0 | 0 | 0 | 0 | 0 | 0 | 1 | 1 | 0 | 0 | 0 | −15.8851 | −15.9 | 0.09 |
| 0 | 0 | 0 | 0 | 0 | 0 | 0 | 0 | 0 | 0 | 0 | 0 | 1 | 1 | 1 | 0 | 0 | 0 | −15.9021 | −15.9 | 0.01 |
| 0 | 0 | 0 | 0 | 0 | 0 | 0 | 0 | 0 | 0 | 0 | 0 | 1 | 1 | 1 | 1 | 0 | 0 | −15.4356 | −15.4 | 0.23 |
| 0 | 0 | 0 | 0 | 0 | 0 | 0 | 0 | 0 | 0 | 0 | 0 | 1 | 1 | 1 | 1 | 1 | 0 | −15.5183 | −15.5 | 0.11 |
| 1 | 1 | 1 | 1 | 1 | 1 | 1 | 1 | 1 | 0 | 1 | 1 | 1 | 1 | 0 | 1 | 0 | 1 | −15.7793 | −15.8 | 0.13 |
| 1 | 0 | 1 | 0 | 1 | 0 | 1 | 0 | 1 | 0 | 1 | 0 | 1 | 0 | 1 | 0 | 1 | 0 | −16.6678 | −16.9 | 1.37 |
| 0 | 1 | 0 | 1 | 0 | 1 | 0 | 1 | 0 | 1 | 0 | 1 | 0 | 1 | 0 | 1 | 0 | 1 | −16.6741 | −16.4 | 1.67 |

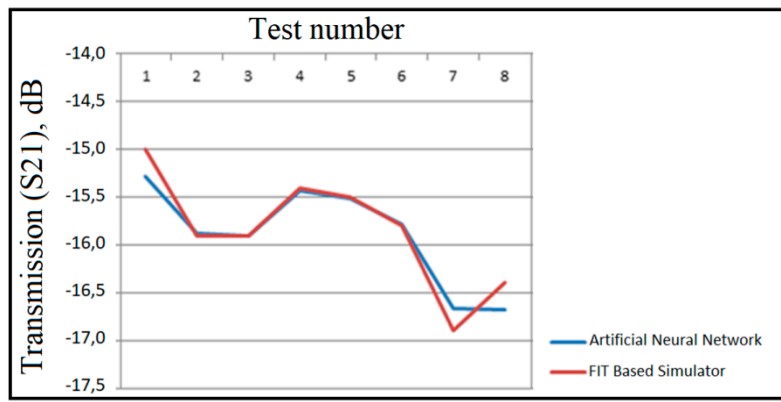

**Figure 5.** Comparison of an artificial neural network and Finite Integration Technique (FIT) based microwave simulator.

Furthermore, random resonator planes were created and imported into the artificial neural network to obtain an optimum design with a reduced mutual coupling effect (S21). The minimum mutual coupling effect obtained by resonator plane is illustrated in Figure 6 and the cross dipole antenna was investigated with and without a resonator plane, which is shown in Figure 7. Moreover, the cross dipole antenna was tested in a microwave simulator with and without a resonator plane and transmission values between each antenna element was obtained as shown in Figure 8. According to the transmission graph, the plane resonator decreases the mutual coupling effect by nearly 2 dBi in the 2.2–2.7 GHz band.

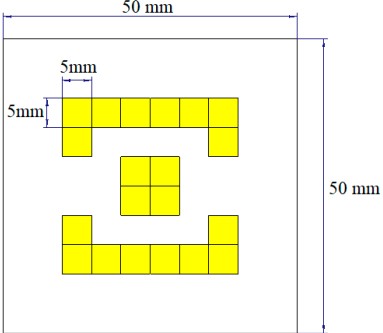

**Figure 6.** A resonator plane designed by a neural network.

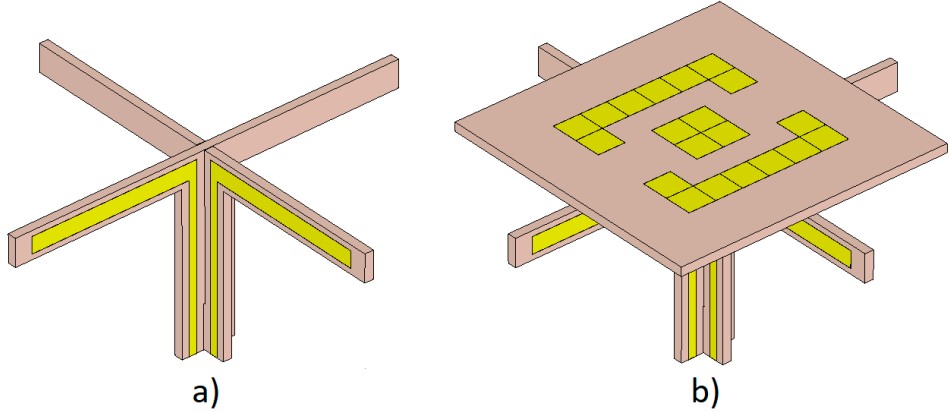

**Figure 7.** Cross dipole antenna (**a**) with a resonator plane and (**b**) without a resonator plane.

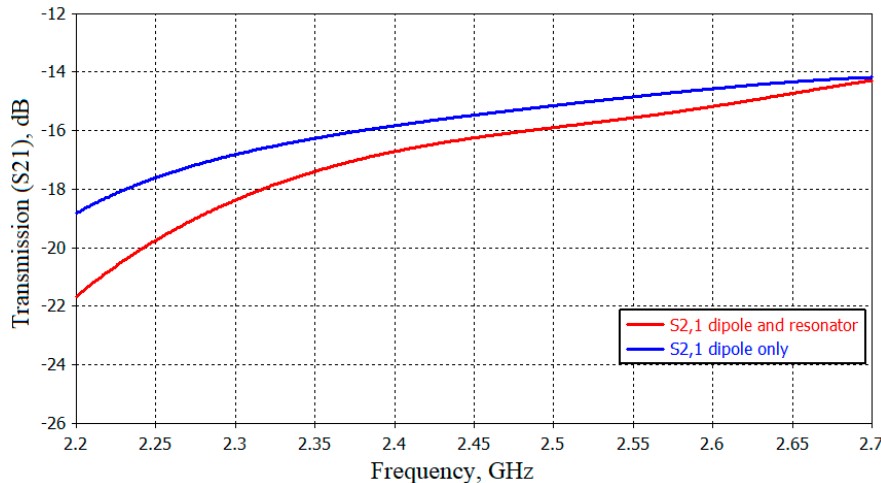

**Figure 8.** The mutual coupling effect between cross dipole antennas.

## 5. Experimental Investigation and Results

The designed cross dipole antenna and plane resonator was fabricated by a LPKF-E33 circuit printer, as shown in Figure 9. The figure shows only a cross dipole antenna with a connected SMA port and plane resonator integrated cross dipole antenna with an SMA port, respectively. Experimental measurements were carried out by an Agilent PNA-L vector network analyzer as given in Figure 10a, and measured transmission values are plotted in Figure 10b. According to the measured transmission values, the resonator plane reduced transmission values between each antenna of cross dipole antenna, which led to a mutual coupling effect being decreased when using this resonator layer. At the frequency of 2.4 GHz, a cross dipole antenna has mutual coupling (S21) of −19.7 dB but it is −25.3 dB with the use of a plane resonator layer. As a result of this study, a trained artificial neural network provides an accurate solution for mutual coupling effect between each element of cross dipole antenna with an error rate of less than 2%, as given in Table 2.

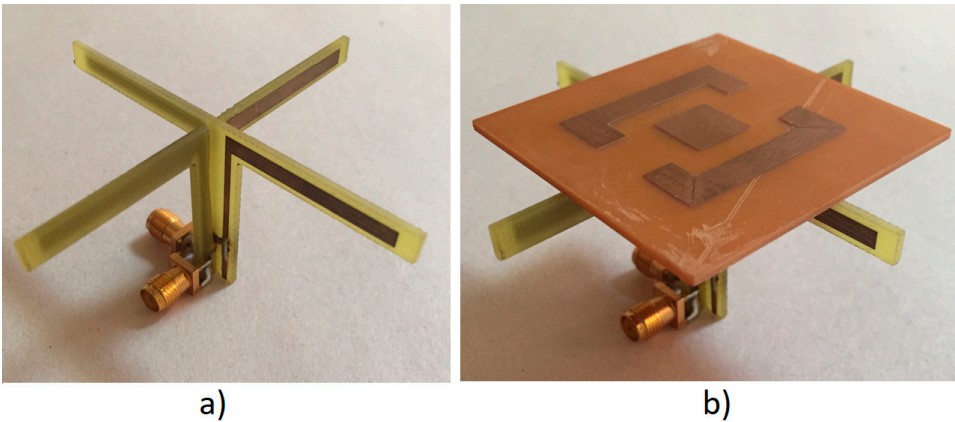

**Figure 9.** (**a**) Fabricated cross dipole antenna and (**b**) resonator layer integrated cross dipole antenna.

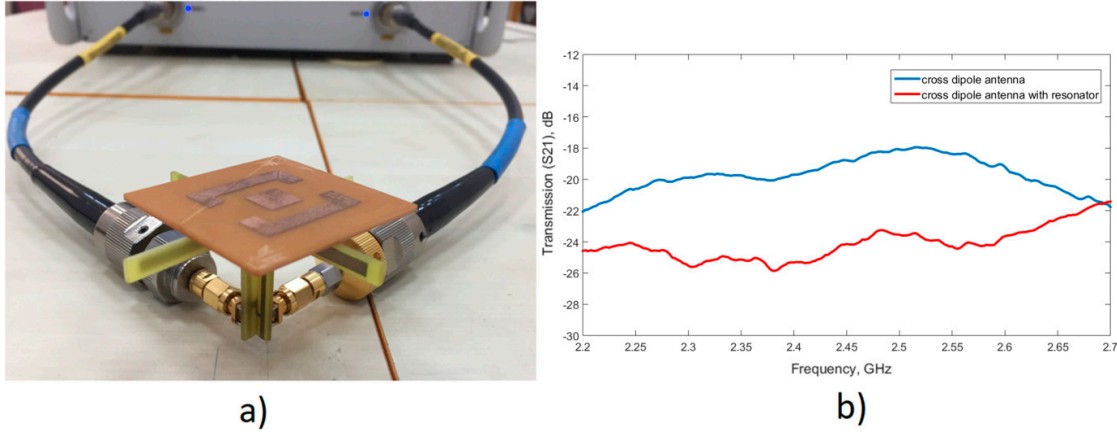

**Figure 10.** (**a**) Experimental measurement and (**b**) transmission graph.

## 6. Conclusions

This research paper presents a resonator layer combined with a cross dipole antenna for wireless communication bands, especially for the 2.2–2.7 GHz band. First of all, a cross dipole antenna was designed in an FIT based microwave simulator. Due to the close distances between each antenna elements constituting the cross dipole antenna, there were −15.6 dB transmissions between each antenna element at 2.4 GHz. Various type resonator layers were then designed and imported into the Matlab environment for neural network processes. Additionally, an artificial neural network was trained by various types of resonators to create a minimum mutual coupling effect between each antenna element of cross dipole antenna. The obtained minimum mutual coupling at 2.4 GHz was −17 dB with a resonator layer. Finally, a resonator layer for a minimum mutual coupling effect and thus for better isolation between antenna elements was fabricated in a microwave laboratory and tested experimentally. The obtained accurate results showed that the designed structure improves the isolation and reduces the transmission values significantly, which is a challenging problem in the antenna industry. In conclusion, it can be easily said that an artificial neural network provides a solution for antenna parameter enhancement applications with different frequency bands.

**Author Contributions:** Conceptualization, E.O. and O.A.; methodology, E.O. and F.O.A.; software, E.O. and F.O.A.; validation, M.K., O.A. and F.O.A.; formal analysis, O.A.; investigation, E.O.; resources, F.O.A.; data curation, M.K.; writing—original draft preparation, E.O.; writing—review and editing, Y.I.A. and M.K.; visualization, E.O.; supervision, O.A. and M.K.; project administration, M.K.; funding acquisition, L.D. All authors have read and agreed to the published version of the manuscript.

**Funding:** This work was supported by the National Key Research and Development Program of China (Grant no. 2017YFA0204600), the National Natural Science Foundation of China (Grant no. 51802352) and the Fundamental Research Funds for the Central Universities of Central South University (Grant no. 2018zzts355).

**Acknowledgments:** The author would like to thanks for Iskenderun Technical University and Central South University for the technical supports.

**Conflicts of Interest:** The authors declare no conflict of interest.

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
