# Peer review of "Mutual Coupling Reduction of Cross-Dipole Antenna for Base Stations by Using a Neural Network Approach"

_applsci, doi:10.3390/app10010378_

Round 1

Reviewer 1 Report

I suggest you revise the last sentence in Abstract reads "one of the obtained results was fabricated".

Section II seems redundant information to me as it is in the title of the paper.

A comparison to other methods in literature and also at different frequencies are missing.

The benefit of using NN over FIT based method is not clearly explained.

Reviewer 2 Report

Paper ID # applsci-676906

Paper title: Mutual Coupling Reduction of Cross-Dipole Antenna for Base Stations by Using Neural Network Approach

Summary and innovation: This article proposed a resonator layer to reduce the mutual coupling effect between each antenna element of a cross dipole antenna. In their design processes, an artificial neural network approach was used for various resonator designs.

General comments: The language needs to be improved in the manuscript. I have found grammatical (especially, clause and determiner) and punctuation errors in the paper. The present format cannot be accepted. It is suggested to revise the manuscript by a professional editor.

Technical comments: The caption was missing from Table 2.

In Figure 5, the titles and units of the X-axis and the Y-axis were missing, respectively.

Regarding Figure 5, the authors claimed the trained neural network was well-matched with the results of FIT based microwave simulator. They did not provide any supportive data to justify their argument. It is essential to provide statistical tests to justify their claim.

They need to demonstrate how much matching and deviation they found between these two results.

Also, what was the standard deviation they obtained from each data point in Figure 5?

Overall assessment: The language needs to be improved. 

The results and discussion section need to address the above-mentioned issues.

Reviewer 3 Report

The authors have studied a resonator layer which reduce mutual coupling effect between each antenna element of cross dipole antenna. The topic in interesting and paper is well written. Reviewer has following minor comments:

The findings in Fig.4. to Fig. 10 are not clearly described. The authors should describe these figures in details. Fig. 3 is blurry.  Please replot it.
